# Photoplethysmography Enabled Wearable Devices and Stress Detection: A Scoping Review

**DOI:** 10.3390/jpm12111792

**Published:** 2022-10-31

**Authors:** Mina Namvari, Jessica Lipoth, Sheida Knight, Ali Akbar Jamali, Mojtaba Hedayati, Raymond J. Spiteri, Shabbir Syed-Abdul

**Affiliations:** 1Department of Computer Science, University of Saskatchewan, Saskatoon, SK S7N 5C9, Canada; 2SUNUM Nanotechnology Research Centre, Sabanci University, Istanbul 34956, Turkey; 3Guilan University of Medical Sciences, Rasht 3363, Guilan, Iran; 4Graduate Institute of Biomedical Informatics, College of Medical Sciences and Technology, Taipei Medical University, Taipei 106, Taiwan; 5International Center for Health Information Technology, College of Medical Science and Technology, Taipei Medical University, Taipei 106, Taiwan; 6School of Gerontology and Long-Term Care, College of Nursing, Taipei Medical University, Taipei 110, Taiwan

**Keywords:** PPG, stress monitoring, wearable devices

## Abstract

Background: Mental and physical health are both important for overall health. Mental health includes emotional, psychological, and social well-being; however, it is often difficult to monitor remotely. The objective of this scoping review is to investigate studies that focus on mental health and stress detection and monitoring using PPG-based wearable sensors. Methods: A literature review for this scoping review was conducted using the PRISMA (Preferred Reporting Items for the Systematic Reviews and Meta-analyses) framework. A total of 290 studies were found in five medical databases (PubMed, Medline, Embase, CINAHL, and Web of Science). Studies were deemed eligible if non-invasive PPG-based wearables were worn on the wrist or ear to measure vital signs of the heart (heart rate, pulse transit time, pulse waves, blood pressure, and blood volume pressure) and analyzed the data qualitatively. Results: Twenty-three studies met the inclusion criteria, with four real-life studies, eighteen clinical studies, and one joint clinical and real-life study. Out of the twenty-three studies, seventeen were published as journal-based articles, and six were conference papers with full texts. Because most of the articles were concerned with physiological and psychological stress, we decided to only include those that focused on stress. In twelve of the twenty articles, a PPG-based sensor alone was used to monitor stress, while in the remaining eight papers, a PPG sensor was used in combination with other sensors. Conclusion: The growing demand for wearable devices for mental health monitoring is evident. However, there is still a significant amount of research required before wearable devices can be used easily and effectively for such monitoring. Although the results of this review indicate that mental health monitoring and stress detection using PPG is possible, there are still many limitations within the current literature, such as a lack of large and diverse studies and ground-truth methods, that need to be addressed before wearable devices can be globally useful to patients.

## 1. Introduction

Mental illness is a common disorder among individuals in most industrialized countries and many emerging economies [1]. The pressure of study, life, and employment can produce a variety of negative emotions that, over the long term, can lead to mental health issues such as chronic stress, anxiety, and depression. Unfortunately, many people do not take the initiative to seek help and consult professionals. Stress is a nearly universal human experience; however, individuals have different ways of defining it. Most people tend to consider stress as a negative emotion, yet this is not always the case. Eustress is described as stress in daily life that has positive connotations such as marriage, promotion, or having a newborn. On the other hand, there is distress, which has negative significance, such as divorce, financial problems, or injury [2]. According to the American Psychological Association [3], there are three types of stress: (1) acute stress, which is short term and not necessarily harmful during which a mild increase in heart rate (HR) and blood pressure (BP) is observed; (2) episodic acute stress, which is when an individual feels stressed frequently, for instance, by over-working regularly (the symptoms are similar yet accumulative and, if poorly managed, can result in heart disease or depression), and (3) chronic stress, which is a result of being in a situation, such as an abusive relationship, war, or natural disaster, that continues for months or years and is the major cause of cardiovascular diseases, severe sleep issues, weight changes, depression, and even suicide.

The mainstream methods of assessing mental health are questionnaires and professional consultation, which are considered expensive, time-consuming, and more importantly, subjective [4]. In addition, the patient may hesitate too long to seek help, and many may never do it. Thus, detection tools to monitor physiological and psychological signals passively, objectively, and continuously are of great importance and interest.

Wearable technology enables a healthcare provider to remotely monitor health parameters and provide more effective care. We recently surveyed the literature on the accuracy of existing PPG-based wearables on health parameters such as heart rate (HR), heart rate variability (HRV) atrial fibrillation (AF), blood pressure (BP), obstructive sleep apnea (OSA), blood glucose (BG), and respiratory rate (RR) [5]. In addition to physiological health, monitoring mental health has gained tremendous attention, specifically over the past decade. The COVID-19 pandemic has further underscored the need for wearable technology in healthcare due to lockdowns, patient unwillingness to leave their homes, and restricted clinical visits. Several reports have documented that the number of people diagnosed with mental illness has risen considerably during the COVID-19 pandemic. However, stress, anxiety, and depression are latent constructs and often difficult to monitor objectively; therefore, there has been a recent focus on the feasibility of creating technology capable of detection of mental states. The autonomic nervous system (ANS) is composed of two branches. The first branch is the sympathetic nervous system (SNS) that drives the “fight or flight” response in stressful situations. The second branch is the parasympathetic nervous system (PNS), which has almost exactly the opposite role of the SNS; it oversees “rest and digestion” [6]. When an individual is exposed to a stressor, changes in HR, BP, RR, heart rate variability (HRV), sweat gland activity, and skin temperature (ST) are observed, all of which are regulated by the ANS [7]. Therefore, continuous monitoring of ANS activity with mobile or wearable devices enables us to capture real-time behavioral and psychosocial data in a precise and confidential manner via monitoring HR, HRV, RR, skin conductance (SC) using photoplethysmography (PPG), electrocardiography (ECG), and galvanic skin response (GSR) sensors. Photoplethysmography, as a potential surrogate to ECG, is an inexpensive, user-friendly, and non-invasive technology that measures the alterations in blood volume via infrared light, thus allowing practical and continuous HR and HRV measurements [8]. Heart rate variability is a promising candidate as a marker of mental health [9,10,11]. Therefore, wearable PPG sensors are great candidates to enable individuals to stay aware of their stress level in real time and manage it. The data collected by various sensors can be analyzed with different techniques. Most of the studies included in this review use machine learning algorithms in the detection of stress using PPG-based wearable devices. Machine Learning (ML) algorithms are promising methods to classify mental health status. This classification is performed considering different features retrieved from data collected by sensors. A list of nomenclature used in this review paper is given in Table 1 as follows.

The objectives of this review are to:Report on recent achievements and advancements in mental health monitoring and stress detection using non-invasive wearable devices equipped with PPG sensors.Identify the existing limitations and gaps in detection of stress using PPG-based wearable devices.Provide direction for future research in this area.

The review is organized as follows. The methods section outlines the process used to select papers for the review, including eligibility criteria and the PRISMA (Preferred Reporting Items for Systematic Reviews and Meta-analyses) diagram. The results section presents data from the included papers; however, because the majority of the papers were based on PPG and stress, they form the focus of this review. The discussion section analyzes the results of the review, identifies limitations in the literature, and makes recommendations for future research.

## 2. Methods

This scoping review was conducted in accordance with PRISMA. Studies were identified by querying the PubMed, Medline, Embase, CINAHL, and Web of Science databases for papers published in the period 2017 to 2022. The goal was to capture eligible studies that used wearable PPG sensors or PPG sensors in combination with other wearables to assess or study mental health.

The following search terms were used in combination in all the databases: (wearables OR smart phone OR smartwatch OR smart device OR monitoring) AND (mental health OR mental disorder OR psychological distress OR mental distress OR stress OR anxiety OR generalized anxiety disorder OR depression OR major depressive disorder) AND (PPG or photoplethysmography). Searches were limited to title, abstract, and keywords. The eligibility criteria were:The trials could be in clinical or real-life environments.Any wearable device type was included.The wearable device used PPG-based technologies to assess mental health, although additional peripheral sensors were permitted.The area of mental health included mental health, mental disorders, psychological distress, stress, anxiety, generalized anxiety disorders, depression, or major depressive disorders.The study was published between January 2017 and April 2021.The study evaluated mental health in humans.The study was published in English.The study was not a review paper.The study was not an abstract paper or conference paper with no full text available.The study did not collect data by placing a finger on a camera.

Five reviewers (M.N., J.L., S.K., A.A.J., M.H.) independently screened the selected articles by their titles and abstracts and evaluated whether a given study met the inclusion criteria. If the information from an abstract was unclear, the full text was retrieved. At all steps of the screening process, the decision to include or exclude an article was made by five reviewers, with M.N. and S.K. serving as the final arbiters.

The extracted data from all the studies were tabulated and reviewed multiple times for their consistency and accuracy. The extracted data included demographic information such as the study type (clinical vs. real-life settings), total number of participants, average age of the population, medical condition of the participants, the stress signal, the stress test, and the methodology used in each article.

## 3. Results

Out of 290 articles screened, 23 papers are included in this review, with 4 real-life studies, 18 clinical studies, and 1 joint real-life and clinical study (Figure 1). Among the 23 articles that fulfill the eligibility criteria, 17 articles are journal papers, and 6 are conference papers with full-text availability. The mental health areas appearing in the included papers are stress (n = 20), depression leading to suicidality (n = 1), PTSD (n = 1), and mental workload (n = 1). Due to the majority of the papers focusing on PPG and stress, we have chosen only to tabulate extracted data from the 20 stress-related studies.

In general, the stress studies included in this review had the same general format. To begin, data were collected from participants using a form of wearable device. To ensure differing levels of stress during the data collection, participants were asked to perform various stress tests. Although many different formal stress tests were used, most tests had the following format: a controlled period of rest followed by an intense activity to induce stress, followed by a controlled rest and recovery period (exceptions to this case are the studies performed in real-life, where the study relied on everyday stressors and periods of rest). Once data were collected, they were processed and classified into categories based on the level of participant stress. Accuracy was then checked against a form of ground-truth method.

Details for all included stress studies can be found in Table 2 and Table 3. Table 2 contains sample size and demographic information for all the included stress studies. Table 3 contains data about the stress detection studies. Additional information is reported in the sub-sections below.

### 3.1. Brief Summary of Anxiety and Depression-Related Papers

There was only one paper was related to detection of suicidal behavior using PPG wearables [31]. In this study, 51 patients presented to the emergency department or admitted to the psychiatric unit for acute suicidality were asked to wear a PPG wrist-worn device for 7 days. The patients were required to complete the Columbia Suicide Severity Scale (CSSRS). An increase in the high-frequency (HF) component of HRV was observed in patients with 25% or more decrease in CSSRS. A study by Cakmak et al. [33] investigated whether a PPG-based research watch could predict PTSD outcomes (e.g., pain, sleep anxiety). Three methods of data collection were applied: patients who wore the watch to collect HRV and actigraphy data, patients who answered a survey, and patients who contributed both to the watch and survey data. The accuracy from the watch was comparable to the survey indicating that PPG-based wearables have potential for PTSD monitoring in long period studies. A study assessed mental workload through HRV data acquired from 19 subjects wearing a Pix Art PPG watch and doing an N-back test [32]. The proposed PPG-based mental workload system performed comparably to the synchronized ECG device measurements.

### 3.2. Stress Detection and PPG

Twelve out of the 20 stress-related studies used only a PPG sensor to monitor stress [7,12,13,14,15,16,17,18,19,20,21,22]. Of these 12 studies, HRV was the most common health parameter. Three separate studies [12,13,15] reported observing the same behavior for HRV from different PPG sensors on different locations: during times of increased stress, HRV decreased. The remaining HRV papers did not specify the behavior of HRV during moments of stress vs. no stress (e.g., if the HRV decreased during moments of stress) but simply reported that they were successfully able to use the parameter to classify moments of stress vs. non stress.

Other health parameters for detecting stress using only PPG included pulse transit time (PTT), pulse waves, and blood volume pressure (BVP). All three parameters were shown to successfully detect stress. However, Celka et al. [7] reported that although changes in pulse waves can be detected during times of stress, those changes are not as obvious as those detected in heart rate (HR), blood pressure (BP), or the ANS. Furthermore, Celka et al. [7] observed that within BP, DBP varied more significantly than SBP; therefore, they suggested the use of DBP to monitor stress. It is also of interest to note that the study using BVP to detect stress [20] was the only study to utilize a database of PPG signals, WESAD [34], instead of collecting their own data.

Studies that utilized solely PPG used two primary locations of sensors: ear [15,19,22,26] and wrist [7,12,13,14,18,23,24,25,27,29]. Sensors located at the ear were less prone to motion artifacts [22]. However, studies using wrist-worn sensors—including a long-term everyday study conducted by [13]—still successfully detected stress, although one of these studies [13] did observe that the wrist-worn PPG signals were more accurate before bed than during the day (i.e., they were more accurate when participants were moving less).

The remaining nine papers used PPG in combination with other sensors [23,24,25,26,27,28,29,30]. Additional sensors included electrodermal activity (EDA)/Galvanic Skin Response (GSR), which detects sweat and skin temperature changes; accelerometers (ACC)/gyroscopes (GYRO), which provide motion information to detect drowsiness and clean artifacts from EDA and PPG signals [28,29]; seismocardiography (SCG), which records cardiac mechanical vibration using an accelerometer-based sensor [35]; and ECG, which measures electrical activity of the heart. In general, studies reported a higher accuracy using sensors in combination with PPG than using PPG alone [24,26,27,30]. One study [25] went as far to determine that stress was detected with higher accuracy when PPG was excluded from the multi-sensor data, although they were quick to conclude that this was likely due to motion artifacts.

Most included studies made use of various commercial products for data acquisition. Devices included the Empatica E4 wristband, Samsung gear wristwatch, James one, and Shimmer 3. All devices successfully detected stress in at least one study; however, there were mixed results. For example, in Can et al. [28], the Empatica E4 sensor detected stress with a higher accuracy than the Samsung gear watch, but in the study by Arsalan et al. [26], the Empatica E4 failed to detect stress entirely.

Any sensor or device that can be connected to WiFi can in principle be part of the Internet of Things (IoT). In the context of this review, sensors such as GSR, ECG, and GYRO can be considered as IoT devices and form part of PPG-based smart systems for stress detection.

### 3.3. Ground-Truth Methods

There are two main categories of ground-truth methods that are used to directly assess a participant’s level of stress: self-report questionnaires and physical-response sampling. Examples of self-report questionnaires include the Perceived Stress Scale (PSS), Stress Self-Rating Scale, NASA-TLX, State-Trait Anxiety Inventory, Visual Analogue Scale (VAS), relative stress scale, Self-Assessment Manikin, and Positive and Negative Affect Schedule test [28]. Such tests involve having a participant answer a series of questions that rate their current perceived experiences and emotions. For physical-response sampling, there are two main methods: testing salivary cortisol [24] and leukocyte measurement [35]; none of the studies in this review used the latter method.

Instead of directly assessing a participant’s stress levels as the ground truth (cortisol test), other studies used additional sensors to either verify the behavior of a certain signal (e.g., HRV) during periods of stress or used a signal that had a known behavior during stress and therefore could reliably indicate periods of stress. An example of the first type of study is [17], in which ECG was used to verify PPG signals. An example of the second type of study is [7], in which BP, a well-known health parameter that increases during periods of stress, was used to determine periods of stress. Five studies did not report any ground-truth methods [14,19,22,26,27].

### 3.4. Common Methodologies for Detecting Stress from PPG

#### 3.4.1. Hardware-Based Methodologies

Photoplethysmography signals can be processed and analyzed by using hardware-based methodologies with wearable devices. Of the twenty papers included in Table 4, twelve papers used some form of hardware-based filter to preprocess or extract the PPG signal [7,12,13,15,16,17,18,21,22,26,27,30]. Three of the papers [18,26,30] used band-pass-type and signal averaging filters to preprocess signals before performing stress classification using ML algorithms. One paper [27] used signal preprocessing before applying a statistical test to the PPG signal.

The remaining eight papers [7,12,13,15,16,17,21,22] used a combination of preprocessing and time/frequency domain analysis to extract RR interval and HR features. For time domain analysis, the most common technique was to determine the derivatives of the PPG signal and use them to determine the peaks and valleys of the signal corresponding to RR-intervals and heart beats [7,12,13,15,16,21,22]. Only one paper [17] used frequency domain analysis in the form of an IIR (infinite impulse response) Bandpass filter combined with sinusoidal modeling to determine the frequency of HR.

#### 3.4.2. Machine Learning-Based Methods

Photoplethysmography signals can also be processed and analyzed using ML. Within the studies included in this review, 13 papers analyzed the data with different ML classification algorithms to classify stress. All papers included feature extraction to improve the performance of the ML algorithms.

In terms of classification algorithms, 9 papers [18,19,20,23,25,26,28,29,33] applied different classifiers on the same datasets, while the remaining 4 papers applied single classification algorithms [14,24,30,32]. The stress status of the participants was divided into varying amounts of classifications across the studies: 8 papers divided them into 2 classes, 3 papers into 3 classes, 1 paper into four classes, and 1 paper used a combination of 2 or 3 classes (see Table 3).

The performance of classification methods was assessed using several evaluation metrics including accuracy, precision, recall, F-measure, and area under the curve (AUC). Because the included papers used different datasets, classifiers, and evaluation metrics, it is not possible to directly compare the performance of the classification algorithms; however, studies that used multiple classification algorithms on the same dataset are amenable to comparing the performance of each algorithm within each study. For instance, [26] performed analytical classification using different ML-based algorithms, including (a) k-Nearest Neighbors (kNN), (b) Support Vector Machine (SVM), (c) Random Forest (RF), (d) Multilayer Perceptron (MLP), and (e) Decision Tree (DT) to predict whether a participant was stressed or not and determined that SVM with RBF kernel achieved the highest precision and recall for all the cases. Moreover, [18] used kNN, SVM, RF, MLP, and XGBoost for binary classification and the F-measure metric to compare different algorithms and reported that RF achieved the best performance among compared methods.

Table 5 summarizes the studies that use ML algorithms to analyze data regarding stress, including the classification algorithm(s) for each study. Moreover, the best performance results for “PPG alone” and “fused PPG with another sensor(s)” are provided.

## 4. Discussion

This section focuses on highlighting gaps or limitations found within the literature and seeks to provide fundamental questions to consider when researching, designing, or using a wearable device to measure mental health, in particular stress detection, within the context of the results of this review. Table 6 refers to a summary of common limitations.

Although this review is aimed at providing an assessment of PPG wearable sensors to detect various mental health issues, most of the papers that met the eligibility criteria focused on stress detection methods. Stress is indeed a prevalent mental health issue; however, there are many other important ones. For example, depression is the leading cause of disability worldwide [36]. The lack of studies on using PPG to evaluate other forms of mental health issues shows a clear gap in the literature. Preliminary studies, such as the one conducted by Sheridan et al. [31], shows that PPG could be greatly beneficial for application in depression—specifically in detecting and intervening during acute suicidality—or for assistance in monitoring PTSD.

The results of this review also raise additional considerations about the state of the technology. The first of these considerations is the design of the devices. Although most studies included in this review generally followed the same framework for experimental set-up (see results section), the devices used to collect PPG data varied greatly—with no two studies using the same device. As is the case in designing a device itself, the specific design choices made by the authors for their studies have corresponding advantages and disadvantages, discussed in detail in the following paragraphs.

One design choice is whether to use only PPG to monitor mental health issues or whether to use a combination of sensors. The results of this review indicate that using a PPG sensor alone to measure HRV is adequate to detect stress; however, results also indicate that adding information from other sensors tends to lead to a higher accuracy of stress detection. Although this may indicate that using multiple sensors instead of only PPG is desirable, limitations arise. The first is the cost: more sensors entail more monetary expense. The second is power consumption: more sensors need more power and hence will limit battery life—which could negatively impact a device meant for continuous monitoring. Time spent charging is time generally lost collecting user data and, as such, should be minimized.

There is also the design choice of processing the PPG data using a hardware-based method or ML. Although it is difficult to compare the accuracy of the two methods in the results of this review due to differing ground truths and datasets, there are still advantages and limitations that can be considered. One advantage in using a hardware-based method is direct access to the physical parameters, such as RR intervals and HR. Unlike with ML, where signals are passed into an algorithm that abstracts the selection of features used for analysis, hardware-based methodologies require deliberate calculation of health parameters to measure stress, potentially allowing for detection of other health states simultaneously with little additional overhead. A second advantage of hardware is speed: dedicated, optimized, and embedded hardware can calculate health parameters much faster than software [37]. However, hardware is much less flexible than ML: [37] it cannot adapt to a specific user, nor can it be easily modified once created. It is also far more monetarily expensive than using ML.

Once a methodology for processing the PPG data is selected, there are still many design choices to be made within the methodology. For hardware, one must decide what type of signal preprocessing to use; for ML, one must decide whether signal preprocessing is desired. Studies included in this review primarily used two types of pre-processing: applying averaging/normalization to the PPG signal (e.g., [7,17,22]) or using filters to remove unwanted signals. Averaging a signal with past input provides an advantage in that it does not remove any of the data; however, smoothing the signals in such a way may cause the device to be slower in detecting changes in mental health because more signals of a new pattern than input signals of an old pattern must be received before the average truly begins to reflect a change. Using a filter to remove noisy signals is likely to be more responsive to changes, provided that cut-off frequencies for the filters are chosen correctly. This is yet another design choice, with each study in this review selecting different cut-off ranges for their devices. Once again, it is difficult to say due to differing datasets and ground-truth methods which cut-offs from the studies are best; however, it is worth considering that cut-off values should be selected to avoid letting in too much noise (as may be the case in study reported by Gurel, N.Z. et al. [30] because the included frequencies are much broader than other papers), while ensuring important signals are not lost (as may be the case with [15], which has a much tighter low pass signal cut-off than most of the other papers). Machine learning also has many design choices to consider including the type of classifier, numbers of layers, activation functions, etc. Although many studies compared multiple types of classifiers as part of the study, the overall results between studies did not agree on a ‘best’ classifier for detecting stress; this indicates that the performance of the classification algorithms could be dataset dependent. Furthermore, when using classifiers, there is also a design choice of how many classifications should be made. The most common classifiers are binary: either stress is detected or not. However, in practice, stress is often experienced in varying levels. Accordingly, it is not clear how many classifications should be used. It is also not clear whether there is an appropriate ground-truth method that can be easily mapped to the classifications to determine whether they are correct.

Aside from methodology, there are design choices to consider in the physical placement of the device. Results of this review indicate that PPG sensors located at the ear were less affected by motion artifacts, leading Tomita et al. [22] to conclude this characteristic made ear-worn PPG sensors more suitable for real-life stress detection [22]. Although the decrease in motion artifacts is a desirable outcome, comfort of the wearer must also be considered. Ear-mounted sensors are non-obtrusive; however, a quick poll of commercial devices shows them to be considerably less popular than wrist-worn devices.

Finally, there is the design choice of whether to build a custom device or whether to use a device that is already commercially available. Once again, both categories have their strengths and weaknesses. Custom wearable devices offer full flexibility in terms of included sensors, data protocols, and the ability to include custom hardware filters. However, they are much more expensive to manufacture. Commercial wearable devices offer less flexibility in terms of available sensors. However, one does not have to invest in developing a new product and may achieve better uptake of a stress detection system if the patient already owns—or is at least familiar with—the wearable device. It is worth noting that none of the studies used the Apple Watch—which is known to have the most accurate PPG sensor [38]. The Fit-Bit, another popular wearable device that contains an SC sensor as well as PPG, was also not used in any of the studies [38]. Studies investigating the potential of these devices for monitoring mental health issues would be an excellent contribution to the literature due to the popularity and accuracy of the two products.

Although there are many considerations revolving around the design of the device itself, one must also take a critical look at the experiments conducted to test the accuracy of these devices. Because all studies in this review followed the same general framework for experimentation, they all suffered similar limitations.

The first limitation within the experiments is accurately assessing when a patient is experiencing stress. Many studies used self-report questionnaires, which rely on a patient’s ability to assess their own experiences. By nature, this assessment is highly subjective and can cause variance in the accuracies depending on the participants. Some studies used physical sampling such as salivary cortisol measurements, which are much less subjective but far more expensive and difficult to use as a measurement tool in everyday life because they require specialized scientific equipment. Other studies measured accuracy simply by comparing the PPG signal—or the health parameter being measured by PPG—to that of another sensor. Although this technically measures the accuracy of the signal or the ability to extract a specific health parameter such as HR, it can be argued that it does not technically measure the accuracy of detecting stress; just because a signal can measure a health parameter such as HR does not mean that the parameter fluctuations indicate stress. In short, a non-subjective, cost-effective solution for assessing the ground truth of a patient’s mental health does not yet appear to be established, making it difficult to confidently draw conclusions on the accuracy of wearable devices in mental health monitoring.

A second limitation within the experiments was the small number of participants. Small participant sample sizes pose an issue for two main reasons: lack of diversity and lack of training data for ML models. Only one study had a participant sample size of greater than 70 (see Table 2). Compared to the thousands of patients suffering from mental health issues, it is highly unlikely that these small participant sample sizes effectively capture the many differing ways in which stress presents itself. Similarly, it is also unlikely that these small sample sizes provided enough data samples to train a generalized ML model for detecting stress. Although the ML models all seemed to be able to succeed in classifying stress for the participants in the study, it is uncertain whether the models would be as successful when attempting to classify someone from outside of the participant group; this is a common risk with ML, although the risk is potentially higher within the included studies due to the small sample sizes used in the training of the models.

Lastly, although the methods for inducing stress in participants were useful to see whether a wearable device could detect stress, they were not reflective of a more real-world setting. The stress-inducing activities used in most of these experiments relied on applying short bursts of extreme stress, followed immediately by rest and relaxation. In everyday life, quickly swinging between these extremes of stress states may not be common, and wearable devices monitoring patients in the real world cannot rely on such clear emotional changes to operate correctly.

Although this review provides the foundations to begin filling in the gaps in the current literature of how to use wearable devices for reliable stress detection, future studies should strive to collect a large number of participants from which to collect data and make them available on a well-documented database for others to analyze or augment. Future studies should also strive to move away from the lab environment and attempt to engineer easy ways for participants to evaluate their mental health as a ground truth, for example by using online applications to remind the users and assess their stress level through questionnaires or by creating at-home stress test kits that participants can self-administer. Studies comparing several technologies with the same dataset may also help contribute to answering the questions about the ‘best’ design choices.

Many of the questions revolving around design choice can in fact be answered by involving prospective patients in the design of the device. Patients suffering from mental health issues are the most reliable source for understanding what is best for them, whether that be input on device size, the comfort for the location of the sensor, the usefulness of any interactive applications, or whether a custom-made device vs. a commercial device would be easier for them to learn to use and make a habit of using. Patients with mental health issues could likely also provide invaluable insights on the experiment design itself, primarily regarding what techniques of data collection would be least intrusive for them—making data collection more reliable. Within this review, no studies mentioned consulting prospective patients as part of the design. Yet, input from the people who will actually be using the device is a simple way to ensure researchers work towards building the best mental health monitoring device possible.

## 5. Conclusions

This scoping review provides an analysis of 23 studies on the usage of wearable PPG sensors for monitoring mental health. The majority of the studies focused on detecting stress. These studies followed similar experimental procedures, but they greatly varied in the design and type of PPG wearables used. Due to the use of different participant sample sizes, the lack of ground truth or the use of subjective ground truth, and the varying methodologies and classifications, conclusions could not be drawn on which device is best in terms of accuracy. However, all studies succeeded in detecting some form of stress-indicating that stress detection with PPG is indeed possible.

Although the current literature shows that the use of PPG wearables to monitor mental health issues is possible, the future challenge will be to determine how to make such monitoring *effective* and *usable* for large-scale, real-world environments. The studies in this review all suffered from limitations to this effect: only having small participant sample sizes for data collection, using orchestrated moments of extreme stress when collecting data, and primarily relying on subjective ground-truth evaluations. To move the technology of wearables forward in mental health monitoring, studies dedicated to collecting PPG data from large and diverse populations and studies focusing on mental health monitoring in real-world settings are crucial. Studies comparing differing technologies and studies incorporating feedback from prospective patients would also contribute to advancing this field. When an individual is under stress or triggered by stressors, the body counteracts by various physiologic and behavioral responses, e.g., the body starts to sweat, skin temperature increases, etc. Studies analyzing the inclusion of other sensors, such as EDA, ECG, or GYRO, could also benefit the current research because the addition of these sensors would allow wearable devices to capture other physical signs of stress such as sweating, temperature increase, or increase in movement.

Monitoring mental health using PPG—or any other sensor—on a wearable device is a new and exciting field that has continued to gain prevalence during the pandemic. The results of the scoping review show that using a wearable device to monitor mental health is possible, but there are still many unanswered questions and gaps in the literature.

## Figures and Tables

**Figure 1 jpm-12-01792-f001:**
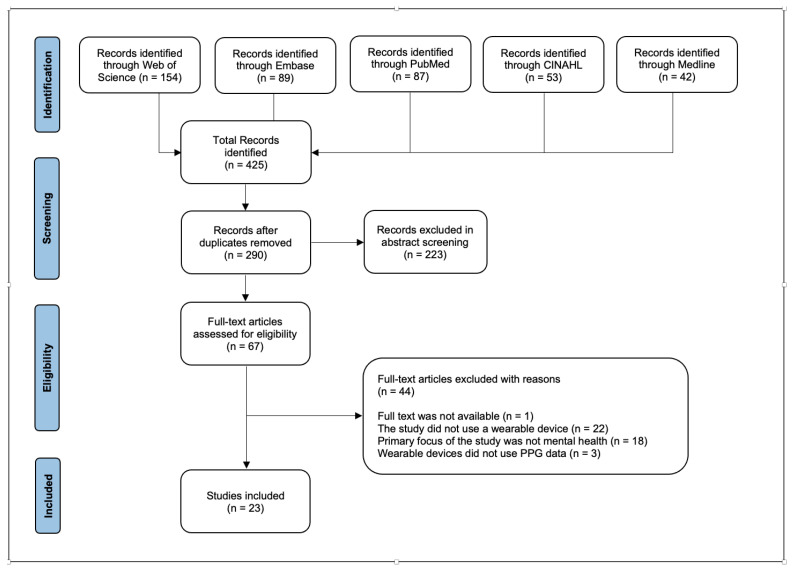
PRISMA flowchart of the identification and selection of studies.

**Table 1 jpm-12-01792-t001:** List of nomenclature used in this paper.

Nomenclature	Referred To	Nomenclature	Referred To
ACC	Accelerometer	PASAT	Paced Auditory Serial Addition Test
ANN	Artificial Neural Network	PAT	Pulse Arrival Time
ANS	Autonomic Nervous System	PCA	Principal Component Analysis
AUC	Area Under the Curve	PEP	Pre-ejection Period
BVP	Blood Volume Pulse	PFC	Prefrontal Cortex
CNN	Convolutional Neural Network	PNS	Parasympathetic nervous system
CSSRS	Columbia Suicide Severity Scale	PP	Perinasal Perspiration
DBP	Diastolic Blood Pressure	PPG	Photoplethysmography
DT	Decision Tree	PRV	Positive Predictive Value
DWT	Discrete Wavelet Transform	PSS	Perceived Stress Scale
ECG	Electrocardiography	PTSD	Post-Traumatic Stress Disorder
EDA	Electrodermal Activity	PTT	Pulse Transit Time
EEG	Electroencephalogram	RF	Random Forest
GSR	Galvanic Skin Response	RR	Respiratory Rate
HF	High Frequency	SBP	Systolic Blood Pressure
HR	Heart Rate	SC	Skin Conductance
HRV	Heart rate Variability	SCG	Seismocardiogram
IAPS	International Affective Picture System	SCWT	Stroop Color-Word Test
IBI	Interbeat Interval	SFFS	Sequential Forward F Selection
kNN	k-Nearest Neighbors	SI	Stress Index
LDA	Linear Discriminant Analysis	SNS	Sympathetic Nervous System
LR	Logistic Regression	ST	Skin Temperature
MDD	Major Depressive Disorder	SVM	Support Vector Machine
ML	Machine Learning	TNR	True Negative Rate

**Table 2 jpm-12-01792-t002:** Basic characteristics of the studies analyzed. The table summarizes the study type, sample size, and average age and the medical condition of the participants.

Study	Study Type	Sample Size	Average Age of Participants (Years)	Medical Condition
[12]	Clinical	12	NR	Healthy
[13]	Real Life	8	21–25	Individuals with no history of cardiac diseases
[14]	Clinical	45	20 to 28	Healthy
[15]	Clinical	18	31.1	Healthy
[16]	Clinical	21	26.3	NR
[17]	Clinical and Real Life	Stress study: 15 Real Life study: 5	NR	NR
[18]	Real Life	14	NR	NR
[19]	Clinical	10	30–58	NR
[20]	Clinical	15	NR (WESAD database)	NR
[21]	Clinical	6	NR (college students)	Healthy
[22]	Clinical	1	A male in his 20′s	NR
[7]	Clinical	10	23–31	Healthy
[23]	Clinical	37–RCDAT dataset	24–27	Healthy
[24]	Clinical	40	73.63	Healthy
[25]	Clinical	control group:17 real-life setting: 1	20–27	NR
[26]	Real Life	40	24.85	Healthy
[27]	Clinical	61	18 to 54 (23.75)	NR
[28]	Real Life	21	NR	NR
[29]	Clinical	26	Males: 20–32 Females: 28–31	Healthy
[30]	Clinical	16	26.7	Healthy
[31]	Clinical	51	13–19	Suicidal adolescent patients
[32]	Clinical	31–CLAS dataset 19–MAUS dataset	NR	NR
[33]	Real Life	1618–AURORA dataset	18–75	Individuals experienced traumatic events

**Table 3 jpm-12-01792-t003:** Stress detection studies in clinical and real-life environments.

Study	Sensor (Location)	Stress Signal	Stress Test	# of Classes	Ground-Truth Method
[12]	PPG (wristband)	HRV	Non-trivial arithmetic task	3 (Baseline, Stress, Recovery)	Perceived stress level (PSL) 0–10
[13]	PPG (wristband)	HRV	Real Life	NR (Stress level)	PSS
[14]	PPG (wristband)	HRV	International Affective Picture System (IAPS)	2 (Distress, Calmness)	NR
[15]	PPG (earlobe)	HRV	Computerized SCWT	2 (Stress, Not-stress)	ECG
[16]	PPG (finger clip)	HRV	Game (python-based program LARA)	2 (Stress, Not-stress)	ECG
[17]	PPG (wristband)	HRV	Modified TSST and Real Life	NA (baseline, Speech, Recovery)	Stress study: ECG Real-life: Polar H7 (ECG)
[18]	PPG (wristwatch)	HRV, HR	Real Life	2 (Stress, Non-stress)	Self-reported questionnaire
[19]	PPG (earlobe)	HRV	Paced Auditory Serial Addition Test (PASAT)	3 (Rest, PASAT, Rest)	NR
[20]	PPG (wristwatch)	BVP	NR	3 (Baseline, Stress, Amusement) 2 (Stress, Non-stress)	WESAD dataset
[21]	PPG (finger clip)	PTT	Modified TSST	NA (Baseline, Speech, Math, Recovery)	ECG
[22]	PPG (earbud)	Pulse Wave	Two-digit addition problems	NA (Stress level)	NR
[7]	PPG (wristband, upper arm, temporal region)	Pulse Wave	Stroop tests	5 (Baseline, Stroop1, Relaxation, Stroop 2, Recovery)	BP and Visual Analogue Scale (VAS) questionnaire
[23]	PPG, GSR (all wristband and finger clip)	HRV, Cardiotach	Audible and visual clips	3 (Relaxing, Normal, Stressful)	Stress state questionnaire on a 0–4 scale
[24]	PPG, EDA, ST (all wristband)	BVP, IBI, EDA, ST	TSST	2 (Stress, Not-stress)	Salivary cortisol measurement
[25]	PPG (wristband), ECG (chest strap), GSR (wristband)	HR, HRV, SC	1- Memory Game, Mosquito Sound, IAPS, Plank, Ice Test, TSST, SCWT 2- Real Life	2 (Stress, Not-stress)	Everyday self-reported stress label
[26]	PPG (earlobe), EEG (headband), GSR (finger clip)	HR, SC, Brain Activity	Real-Life Public Speaking	2 (Stress, Not-stress)	NR
[27]	PPG (wristband), ECG (chest strap), EDA (wristband)	HR, EDA, RR, PP (thermal camera)	Computer task: Essay Writing, Calming Video/Stroop Test, Dual task, Online Presentation	NR	NR
[28]	PPG, EDA, ACC (all wristwatch)	HR, SC, ST, ACC	Real Life	3 (Low stress, Medium stress, High stress)	NASA-TLX and self-reported questionnaires
[29]	PPG, GSR, ST, ACC, GYRO (all wristband)	HRV, ST, ACC	Simulated Indoor Driving Environment	4 (Normal, Stress, Fatigue, Drowsiness)	Self-reported Stress feedback on a 1–5 scale
[30]	PPG (headband), ECG (chest and hips), SCG (chest)	HR, PEP, PTT, PAT PPG amplitude PFC Oxygenation Markers	Mental arithmetic tasks, N-back memory tasks	3 (Easy, Medium, Hard)	NASA-TLX questionnaire

**Table 4 jpm-12-01792-t004:** Hardware-based methodologies for detecting stress from PPG signals.

Study	Signal Pre-Processing	Stress Detection Methodology
[12]	Butterworth low pass filter (2 Hz). Distribution filter, threshold detection	RR interval detection with time domain analysis
[30]	Finite Impulse Bandpass (0.8 Hz- 10 Hz)	Machine Learning
[15]	Low pass filter (5 Hz). Linear extrapolation based on rolling average	RR interval detection with time domain analysis
[18]	Butterband band pass (0.7 Hz–3.5 Hz); moving average filter	Machine Learning
[16]	Cubic spline interpolation on clipped signals, Savitsky-Golay filter	RR interval detection with time domain analysis
[26]	Savitsky-Golay filter	Support Vector Machine
[7]	Moving average filter	RR interval detection with time domain analysis
[27]	Normalization via subtracting baseline signal	T-test
[13]	3 stage band pass (0.5 Hz–11 Hz, 0.8 Hz–3 Hz, 0.9 Hz–1.6 Hz)	Sliding window RR detection with time domain analysis
[22]	Moving average filter	Sliding window PPG and PPG velocity signal analysis
[21]	None	RR interval detection
[17]	Noise removal via least mean squares	HR frequency analysis using IIR Bandpass filter and sinusoidal modeling.

**Table 5 jpm-12-01792-t005:** Stress detection studies using ML-based approaches.

Study	Methods	Classification Classes	Best Performance
[23]	1- Normalization 2- Feature extraction 3- Feature selection using: (A) Alpha-investing (B) OSFS (C) MRMR (D) Chi-square 4- Classification using: (A) kNN (B) MLP ANN (C) Naïve Bayes (D) SVM	1- Stressful 2- Relaxing 3- Normal	PPG ONLY: (kNN = 5 with Chi-square, Accuracy = 80.74) FUSED PPG: PPG + GSR: (kNN = 3 with Chi-square, Accuracy = 85.03)
[29]	1- Normalization using 5 methods 2- Feature construction 3- Feature extraction using: (A) ANOVA (B) SFFS 4- Classification using: (A) MLP (B) SVM + WTA (C) SVM + MWV (D) Naive Bayes	1- Normal 2- Stress 3- Fatigue 4- Drowsiness	PPG ONLY: MLP (Accuracy = 98.43)
[33]	1- Feature extraction: 2- Classification using: (A) SVM (B) LR (C) MLP	1- PROM-Pain4a ≥ 66.6 and PCL-5 ≥ 31 2- PROM-Pain4a < 55.6 and PCL-5 < 31	PPG ONLY PTSD: (A) Linear SVM (AUC = 0.73 ± 0.03) (B) LR (AUC = 0.73 ± 0.03) (C) MLP (AUC = 0.73 ± 0.03) PTSD Sleep Anx./Panic: MLP (AUC = 0.79 ± 0.05) PTSD Pain Int.: Linear SVM (AUC = 0.76 ± 0.04)
[26]	1- Feature construction 2- Feature extraction using: (A) Wrapper method 3- Classification using: (A) kNN (B) SVM (C) RF (D) MLP (E) DT	1- Not stressed 2- Stressed	PPG ONLY: SVM (Accuracy = 80.00, f-measure = 0.79, Kappa Values = 0.72) FUSED PPG: SVM (Accuracy = 96.25, f-measure = 0.96, Kappa Values = 0.87)
[18]	1- Feature extraction 2- Feature selection 3- Classification using: (A) kNN (B) SVM (C) RF (D) MLP (E) XGBoost	1- Not stressed 2- Stressed	PPG ONLY: 1- A little bit VS. baseline MLP (F-measure = 0.73 ± 0.06) 2- Some VS. baseline Random Forest (RF) (F-measure = 0.71 ± 0.05) 3- Extremely VS. baseline: RF (F-measure = 0.76 ± 0.05) 4- Some, a lot or extremely VS. a little bit or not at all XGBoost (F-measure = 0.63 ± 0.04)
[14]	1- Feature Extraction 2- Classification using: (A) DT	1- Distress 2- Calmness	PPG ONLY DT (AUC = 0.75)
[19]	1- Feature extraction 2- Feature selection 3- Classification using: (A) AdaBoost + 11 other classifiers	1- Not stressed 2- Stressed	PPG ONLY AdaBoost (Accuracy = 0.93, Precision = 0.93)
[32]	1- Feature extraction 2- Feature selection 3- Relationship between stress or drowsiness using: (A) SVM	1- High MW state 2- Low MW state	PPG ONLY SVM (Accuracy = 0.78)
[20]	1- Feature extraction 2- Feature selection: (A) Z-score normalization 3- Classification using: (A) CNN (B) Hybrid CNN	1- class classification: (A) Baseline (B) Stress (C) Amusement 2-class classification: (A) Not stressed (B) Stressed	PPG ONLY Two-class Hybrid CNN (Accuracy = 0.89, F-measure = 0.87) Three-class Hybrid CNN (Accuracy = 0.75, F-measure = 0.64)
[28]	1- Feature Extraction (A) HRV (B) ACC (C) EDA 2- Classification using: (A) PCA + LDA (B) PCA + SVM (radial) (C) kNN (D) LR (E) RF (F) MLP	1- Stress Level 1 (index: 0–30) 2– Stress Level 2 (index: 35–75) 3– stress Level 3 (index: 80–100)	FUSED PPG: HR + accelerometer for Empatica E4 MLP (Accuracy = 92.19, f-Measure = 90.3, Precision = 91.4, Recall = 90.2)
[30]	1- Feature extraction 2- Feature selection (A) ANOVA (B) Benjamini-Hochberg Procedure 3- Classification using: (A) RF	1- Rest 2- Arithmetic 3- N-back task	FUSED PPG: PPG + ECG + SCG RF (Accuracy = 0.85 ± 0.09, Recall = 0.84 ± 0.14, Precision = 0.83 ± 0.1, F1 = 0.80 ± 0.13)
[25]	1- Feature extraction 2- Feature selection (A) Greedy Stepwise Method 3- Classification using: (A) kNN (B) SVM (C) Naïve Bayes	1- Not stressed 2- Stressed	FUSED PPG: PPG + ECG + GSR SVM (Accuracy = 0.95)
[24]	(I) WRISTBAND BASED STRESS DETECTION 1- Feature extraction 2- Feature selection (A) Supervised feature selection 3- Classification using: (A) Random Forest (II) BLOOD PRESSURE ESTIMATION 1- Feature extraction 2- Feature selection (A) Supervised feature selection 3- Regression studies using: (A) MLP (B) DT (C) Adaboost (D) Adaboost + DT	1- Not stressed 2- Stressed	FUSED PPG: PPG + EDA RF (Accuracy = 0.94, F-measure = 0.92)

**Table 6 jpm-12-01792-t006:** Limitations of PPG stress-detection.

Category	Limitations
Experimental Limitations Participants Data collection	(1) Small participant sample sizes (2) Relatively homogenous population (3) Failure to report participants’ medical history or additional conditions (4) Exclusion of participants with health conditions (5) Lack of a ground-truth method (6) Use of subjective stress evaluations (7) Poorly designed stress induction process (8) Non-automated process for noisy signal processing (9) Neglected to account for motion and motion artifacts
Device Design Physical Device Methodologies	(1) Device was obtrusive (2) Lack of built-in battery; the device did not have an integrated battery (3) Lack of accompanying user-friendly app to communicate data (4) Lack of consistent cut off values for filters (5) The absence of commonly used commercial devices (e.g., Apple Watch and FitBit) among the studies (6) Use of small datasets that cannot reliably train a generalized ML model (7) Lack of continuum measurement of stress level

## Data Availability

Not applicable.

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
