# Peer review of "Photoplethysmography Enabled Wearable Devices and Stress Detection: A Scoping Review"

_jpm, 2022, doi:10.3390/jpm12111792_

Round 1

Reviewer 1 Report

Review of the manuscript ID: jpm-1971206: Photoplethysmography Enabled Wearable Devices and Stress Detection: A Scoping Review

The authors reviewed studies on the usage of wearable PPG sensors for monitoring mental health. This study was probably motivated by the following observations: the lack of methodology of the different published trials and the absence of fully convincing conclusions. The authors used a good approach to analyze the different studies, and the different tables are useful to understand the approaches used in each trial and to evidence the weakness of the stress classification. This paper is of certain interest to the reader and would probably motivate researchers to strengthen their design of the experiment. I have only a few recommendations listed hereafter, mainly about the discussion section. Also, even though PPG is a well-known method, I recommend adding typical device schematics and PPG signals to illustrate the different signal processing methods described in the present manuscript.

Discussion:

 Table 6 is very instructive. For item 1 (sample size) – do the authors of the different trials specify the rationale for the sample size? It is usually something that is evaded. For item 7 (bottom line) – binary classification – the classification must be consistent with the one used to evaluate independently of the PPG analysis the stress level (ideally using physical measurements like cortisol, but self-tests could also provide a better picture (although subjective) of the stress level than a minimalist binary scale.

 The term “lastly” is used several times in the discussion, please revise.

Direction for future research:

There is probably a correlation between stress and specific trends in the PPG signal. However, as suggested by some studies, additional sensors may be useful to refine the diagnosis or confirm the origin of this specific trend. In the “direction for future research”, it could be great to elaborate more on this point because it is not yet proven that one single sensor can provide reliable stress detection in real-life conditions, as you suggested in this review.

Author Response

Responses to Reviewer #1 comments:

The authors reviewed studies on the usage of wearable PPG sensors for monitoring mental health. This study was probably motivated by the following observations: the lack of methodology of the different published trials and the absence of fully convincing conclusions. The authors used a good approach to analyze the different studies, and the different tables are useful to understand the approaches used in each trial and to evidence the weakness of the stress classification. This paper is of certain interest to the reader and would probably motivate researchers to strengthen their design of the experiment. I have only a few recommendations listed hereafter, mainly about the discussion section. Also, even though PPG is a well-known method, I recommend adding typical device schematics and PPG signals to illustrate the different signal processing methods described in the present manuscript.

We thank the referee for expressing their interest and valuable comment. The locations of the sensors are added to column 2 of Table 3. The acquired signals can be found in column 3 of Table 3. Because each study reported different signal processing methods, we cannot fully describe all the methods in this manuscript. We focused on the results of the analysis, i.e., whether PPG can detect stress accurately or not. 

Discussion:

 Table 6 is very instructive. For item 1 (sample size) – do the authors of the different trials specify the rationale for the sample size? 

We thank the referee for the comment. The studies did not report any specific standard to choose the sample size. Aside from the criteria they had for recruiting, the authors did not provide any specific reason for choosing the size of population. For item 7 (bottom line) – binary classification – the classification must be consistent with the one used to evaluate independently of the PPG analysis the stress level (ideally using physical measurements like cortisol, but self-tests could also provide a better picture (although subjective) of the stress level than a minimalist binary scale.

Item 7 of the third row of Table 6 was changed to “Lack of continuum measurement of stress level” to reflect the reviewer’s comment.

 The term “lastly” is used several times in the discussion, please revise.

We thank the referee for the sharp eye; “lastly” on page 21 was changed to “finally”.

Reviewer 2 Report

In this paper, the authors have presented a detailed review on Photoplethysmography Enabled Wearable Devices and Stress Detection. The paper has several flaws and the following suggestions need to be incorporated to improve its quality.

1. It could be nice if authors can write the details regarding the PPG signal databases.

2. mathematical details regarding various signal processing-based methods for PPG data analysis must be added.

3. Please mention the details regarding the recording devices for PPG data acquisition. like wrist PPG , fingertip PPG, etc

4. Please add the description regarding the association of IoT and PPG-based smart systems for stress detection.

5. A paragraph regarding the selection of PPG over other biosignals such as ECG and heart sounds must be added for stress recognition.

6. The paper must be corrected by a native English speaker before revision.

Author Response

Responses to Reviewer #2 comments:

There is probably a correlation between stress and specific trends in the PPG signal. However, as suggested by some studies, additional sensors may be useful to refine the diagnosis or confirm the origin of this specific trend. In the “direction for future research”, it could be great to elaborate more on this point because it is not yet proven that one single sensor can provide reliable stress detection in real-life conditions, as you suggested in this review.

Thanks to the referee for the comment and agreeing with our results. Stress cannot be measured objectively/directly like blood pressure, heart rate, blood sugar, etc. Thus, to obtain more accuracy in stress detection, various sensors are usually used in combination. This was added to the conclusion on P. 22 and 23: When an individual is under stress or triggered by stressors, the body counteracts by various physiologic and behavioral responses, e.g., the body starts to sweat, skin temperature increases, etc. Studies analyzing the inclusion of other sensors, such as EDA, ECG, or GYRO, could also benefit the current research because the addition of these sensors would allow wearable devices to capture other physical signs of stress such as sweating, temperature increase, or increase in movement.

Comments and Suggestions for Authors

In this paper, the authors have presented a detailed review on Photoplethysmography Enabled Wearable Devices and Stress Detection. The paper has several flaws and the following suggestions need to be incorporated to improve its quality.

  1. It could be nice if authors can write the details regarding the PPG signal databases.

We thank the referee for their comment. In this review, we aimed to investigate whether PPG signals can be utilized to detect and measure stress. Each study has collected PPG signals using various wearable devices. Unfortunately, we do not have access to the databases for these studies.

  1. mathematical details regarding various signal processing-based methods for PPG data analysis must be added.

We thank the referee for their comment. In this review, we aimed to explore whether PPG signals can be utilized to detect and measure stress. For this purpose, we also discussed which methods were used in each study along with their accuracy. We have omitted the mathematical details because they would significantly add to the length of the paper likely without adding significant value to the intended audience.

  1. Please mention the details regarding the recording devices for PPG data acquisition. like wrist PPG , fingertip PPG, etc

We thank the referee for their comment. The required details were added to column 2 of Table 3.

  1. Please add the description regarding the association of IoT and PPG-based smart systems for stress detection.

Thank you for mentioning this, we have added the following paragraph to P. 11 in the manuscript.

Any sensor or device that can be connected to WiFi can in principle be part of the Internet of Things (IoT). In the context of this review, sensors such as GSR, ECG, and GYRO can be considered as IoT devices and form part of PPG-based smart systems for stress detection.

  1. A paragraph regarding the selection of PPG over other biosignals such as ECG and heart sounds must be added for stress recognition.

We thank the referee for their comment. In the second paragraph of P. 11, we have talked about the combination of other sensors that are used with PPG in stress detection among the included studies. 

  1. The paper must be corrected by a native English speaker before revision.

Done.

Round 2

Reviewer 2 Report

All comments are correctly addressed.